# Adaptive UAV Navigation Method Based on AHRS

**DOI:** 10.3390/s24082518

**Published:** 2024-04-14

**Authors:** Yin Lu, Zhipeng Li, Jun Xiong, Ke Lv

**Affiliations:** 1School of Internet of Things, Nanjing University of Posts and Telecommunications, Nanjing 210003, China; xiongjun@njupt.edu.cn; 2Key Lab of Broadband Wireless Communication and Sensor Network Technology, Nanjing University of Posts and Telecommunications, Ministry of Education, Nanjing 210003, China; 3School of Communication and Information Engineering, Nanjing University of Posts and Telecommunications, Nanjing 210003, China; 1221014505@njupt.edu.cn (Z.L.); lwtg2009@163.com (K.L.)

**Keywords:** Unmanned Aerial Vehicle relative navigation, Attitude and heading reference system, Kalman filter, Tasmanian devil optimization

## Abstract

To address the inaccuracy of the Constant Acceleration/Constant Velocity (CA/CV) model as the state equation in describing the relative motion state in UAV relative navigation, an adaptive UAV relative navigation method is proposed, which is based on the UAV attitude information provided by Attitude and Heading Reference System (AHRS). The proposed method utilizes the AHRS output attitude parameters as the benchmark for dead reckoning and derives a relative navigation state equation with attitude error as process noise. By integrating the extended Kalman filter output for relative state estimation and employing an adaptive decision rule designed using the innovation of the filter update phase, the proposed method recalculates motion states deviating from the actual motion using the Tasmanian Devil Optimization (TDO) algorithm. The simulation results show that, compared with the CA/CV model, the proposed method reduces the relative position errors by 12%, 23%, and 32% in the X, Y, and Z directions, respectively, and that it reduces the relative velocity errors by 350%, 330%, and 300%, respectively. There is a significant improvement in the relative navigation accuracy.

## 1. Introduction

Unmanned Aerial Vehicle (UAV) relative navigation is achieved by measuring elements to obtain data such as orientation and distance of UAV formations, determining the relative attitude, position, and velocity of UAVs within the formation. This technology is widely applied in various scenarios, including formation flying, aerial refueling, and autonomous rendezvous and docking of spacecraft. Currently, the more common method for relative navigation involves the combination of the Global Navigation Satellite System (GNSS) and Inertial Navigation System (INS), which provides high precision in relative positioning by obtaining the UAV’s location information. However, with the continuous advancement in UAV technology, higher demands are placed on the precision of relative navigation.

In terms of the relative motion model, many studies have proposed different designs for relative motion state equations. Traditional UAV relative motion estimation usually adopts the CA/CV model [1], which neglects the issue of relative attitudes between UAVs, further affecting the calculation precision of other relative navigation states like position and velocity. Adam and John proposed relative attitude motion equations and relative barycenter motion equations based on attitude quaternions, relative velocity, and relative position as state variables to describe UAV relative motion. However, the complex parameter calculations of this method have some impact on algorithm efficiency [2]. Cheng, J et al. considered the characteristics of close-range accompanying flight of UAV swarms. By omitting the elevation dimension of the navigation coordinate system and assuming that all UAVs are in the same local navigation coordinate system, the H coordinate system is proposed [3]. Based on this system, UAV relative motion equations are derived, simplifying the traditional relative barycenter equation’s parameter calculations and improving algorithm efficiency. However, as it replaces the coordinate system transformation’s three rotations with a single rotation, further improvements in system accuracy are needed. Aiming at the problem of UAV landing on the ship’s body, Zewei Zheng, Zhenghao Jin, et al. proposed a relative model, which was based on the six-degrees-of-freedom (6-DOF) UAV and carrier model to establish a coupled six-degrees-of-freedom nonlinear relative motion model. However, due to the slow change in the motion of the ship’s body relative to the UAV, the study neglected the effect of yaw angle in the relative motion [4].

As UAV clusters are widely used, UAV cooperative navigation methods are required to be able to further improve their solving performance. Reference [1] proposed a relative navigation method based on ultra-wideband/relative difference, utilizing measurements from multiple sensors such as GNSS/INS tight combination positioning data, ultra-wideband (UWB) ranging, and INS outputs. Subsequently, extended Kalman filtering is used for data fusion, reducing the error in relative navigation state quantities. However, due to the high maneuverability of UAVs, the traditional Kalman filtering method cannot adapt to the sudden changes in the motion state of UAVs. Under extreme conditions, the method even causes its output to diverge [5,6,7,8]. To solve this problem, there are two main methods: (1) The system state equations are derived by improving the accuracy and adaptability of the motion model to describe the motion state. Alternatively, the accuracy of the observation equation modeling is improved by comprehensively considering the observation noise of the sensors. (2) Redesigning the filtering method to enhance the filtering method’s ability to deal with the problem when it occurs in the system, and to improve the overall performance of the system. The previous introduction of relative states of motion belongs to Method 1. Reference [9] devises adaptive filtering centered on innovation during the filtering process. The preferred method within this category is the strong tracking filter (STF) proposed in Reference [10]. The STF enhances the system’s adaptability by computing the asymptotic cancellation factor. This factor ensures that the sequence of residuals becomes orthogonal, thereby compensating for significant prediction errors [11,12,13]. The theoretical derivation of constructing STF is more complicated, and there are problems such as large computational volume for solving the asymptotic cancellation factor and arbitrary insertion position [14]. Reference [15] proposed a novel adaptive Sage-Husa, which solves divergence caused by wrong selection values and has strong robustness. Reference [16] studies a joint filtering algorithm based on multi-source sensors. The algorithmic framework consists of a main filter and multiple sub-filters. These sub-filters estimate and output the main filter interdependently to obtain a globally optimal solution. However, all these methods use the extended Kalman filter (EKF). This filter is linearized using a Taylor series expansion. It omits the higher-order terms of the nonlinearization and is less stable in the case of strongly nonlinear equations of state. Arasaratnam proposed the Cubature Kalman Filter (CKF) algorithm, which uses the spherical radial volume criterion to approximate the optimal estimation of the state’s posterior distribution [17,18]. Based on CKF, Reference [19] proposed an efficient adaptive filtering algorithm. The algorithm calculates innovative values and sets decision rules by observing equations and sensor observations. Compensation is performed when the state equations do not accurately describe the system, thus improving navigation accuracy. However, the setting of compensation coefficients in this method depends on empirical values and cannot be applied to different navigation scenarios.

Despite the simplicity of Kalman filter-based collaborative navigation methods, there are two significant drawbacks of such methods: (1) they do not apply to large-scale clusters, and the excessive number of collaborative nodes will increase the computation amount of matrix multiplication and inverse computation without any limitation; and (2) There is a certain performance loss, and they cannot make full use of geometric constraints between nodes to further improve the estimation performance. Collaborative navigation techniques based on optimization algorithms can solve the above problems. Combined with GNSS/UWB systems through optimization algorithms, Günther Retscher et al. propose the fusion of GNSS pseudoranges with UWB ranges based on clustering and Weighted Least Squares (WLS) [20]. Recently, nature-inspired methods have become increasingly popular in UAV navigation because of their ability to efficiently deal with dynamic constraints due to their effectiveness in handling UAVs and searching for dynamic constraints of UAVs. These include Cuckoo Search, Genetic Algorithm (GE), Differential Evolutionary Algorithm (DE), Artificial Bee Colony Algorithm (ABC), Ant Colony Optimization Algorithm (ACO), Ant Colony Optimization (ACO), and Particle Swarm Optimization (PSO) [21,22,23,24,25,26,27].

In response to the challenges and research deficiencies mentioned above, this paper considers the attitude changes during UAV relative motion. To enhance the precision and stability of the UAV relative navigation system, an adaptive UAV navigation method based on AHRS is proposed. The main research work and contributions of this paper are as follows:An AHRS-based novel UAV relative motion model is proposed. The model describes the relative motion using the specific force equation with the UAV attitude output from the AHRS. The model fully takes into account the effect of relative attitude on relative velocity. Therefore, compared with the CA/CV model, this model can improve the overall positioning accuracy of the relative positioning method. In addition, the model uses AHRS as the solution system for the UAV attitude. While ensuring the accuracy of parameter precision, it can realize the correction of attitude error. Compared with the existing relative model, this innovative model reduces the computational complexity of the model for modeling the relative error, while improving the accuracy.A TDO-based adaptive filtering method is proposed. The method utilizes the innovation in the EKF process to design the adaptive judgment rule. This method solves the problem of increased error in Kalman filter state estimation due to abrupt changes during relative motion. Meanwhile, the objective function of the optimization algorithm is constructed using multi-sensor conditions such as AHRS, UWB, and GNSS. A new TDO algorithm is also used to correct the estimates that deviate from the true motion state during the filtering process. Finally, the optimization algorithm is combined with the filtering algorithm to form the TDO adaptive filtering algorithm. The algorithm has better convergence and accuracy than traditional optimization algorithms for high-dimensional optimization problems such as UAV positioning.The performance of the TDO algorithm is compared with other traditional optimization algorithms by performing simulation verification. According to the results, the TDO algorithm has good stability and accuracy when dealing with the problem of UAV scenarios. Meanwhile, by comparing the method proposed in this paper with the traditional relative localization methods, it can be obtained that the new relative localization model can better deal with the relative motion state. Moreover, the TDO adaptive filtering algorithm can improve the accuracy of the method by correcting the deviation from the real motion trajectory.

The rest of this paper is organized as follows. Section 2 describes the system model. Section 3 provides a detailed description of the relative positioning method. Section 4 and Section 5 present the simulation results and conclude the paper.

## 2. System Model

The traditional CA/CV model does not sufficiently consider the relative attitude changes between UAVs, resulting in the system model’s imprecise description of the relative motion state. This inadequacy leads to increased solution errors or even divergence in practical applications. Considering the need to use UAV attitude parameters to construct the relative navigation state equation, the AHRS is introduced. The AHRS employs a triaxial magnetometer, a triaxial gyroscope, and a triaxial accelerometer. It describes the attitude of the moving body using quaternions, measures angular velocity through the gyroscope, and integrates accelerometer and magnetometer data using the Kalman filter algorithm to correct the attitude quaternions, thereby enhancing the accuracy of attitude determination [28]. The structure of the AHRS attitude determination system is shown in Figure 1. Here, the quaternion ***q*** and gyroscope errors are taken as state variables, and the magnetometer output ***m^b^*** and accelerometer output ***a^b^*** serve as observational values in the Kalman filter measurement update. Finally, after the measurement update, gyroscope error ***e*** is used for real-time correction of the gyroscope, and the relationship between quaternion ***q*** and the direction cosine matrix is utilized to obtain the UAV’s attitude information.

The UAV utilizes the output from the AHRS system as the real-time body attitude angles and employs these angles to obtain the coordinate transformation matrix. The transformation relationship is as follows:(1)$$C_{n}^{b}=\left[\begin{array}{ccc} 1 & 0 & 0\\ 0 & \cos\varphi & \sin\varphi \\ 0 & -\sin\varphi & \cos\varphi \end{array}\right]\left[\begin{array}{ccc} \cos\theta & 0 & -\sin\theta \\ 0 & 1 & 0\\ \sin\theta & 0 & \cos\theta \end{array}\right]\left[\begin{array}{ccc} \cos\psi & \sin\psi & 0\\ -\sin\psi & \cos\psi & 0\\ 0 & 0 & 1 \end{array}\right]$$
where *φ* is the roll angle, *θ* is the pitch angle, and *ψ* is the yaw angle. This transformation achieves the conversion between the body coordinate system and the navigation coordinate system.

The specific force *f_b_* is the theoretical output of the accelerometer. Its output is affected by gravity. Based on the definition of specific force and neglecting minor errors, the output of the UAV accelerometer is expressed as [29,30]:(2)$$f_{b}=C_{n}^{b}\left(a_{n}+g_{n}\right)+\delta f$$
where ***a_n_*** is the acceleration in the navigation coordinate system, ***g_n_*** is the projection of the gravitational vector in the navigation coordinate system, and δf is the acceleration error. Because of the derivative relationship between acceleration and velocity, the UAV velocity differential equation is obtained according to Equation (2). ${\overset{\cdot }{v}}_{n}$ is expressed as:(3)$${\overset{\cdot }{v}}_{n}=a_{n}=C_{b}^{n}f_{b}-g_{n}+C_{b}^{n}\delta f_{\varepsilon }$$

Considering that the AHRS system output attitude angle error can affect the whole system, the transfer of the *b* coordinate system to the N-coordinate system is viewed as two coordinate system rotations. $C_{b}^{b^{e}}$ consists of the attitude error of the AHRS.
(4)$$\begin{array}{c} C_{b}^{n}=C_{b^{e}}^{n}C_{b}^{b^{e}}\\ C_{b}^{b^{e}}=\left[\begin{array}{ccc} 1 & 0 & 0\\ 0 & \cos\delta \varphi & \sin\delta \varphi \\ 0 & -\sin\delta \varphi & \cos\delta \varphi \end{array}\right]\left[\begin{array}{ccc} \cos\delta \theta & 0 & -\sin\delta \theta \\ 0 & 1 & 0\\ \sin\delta \theta & 0 & \cos\delta \theta \end{array}\right]\left[\begin{array}{ccc} \cos\delta \psi & \sin\delta \psi & 0\\ -\sin\delta \psi & \cos\delta \psi & 0\\ 0 & 0 & 1 \end{array}\right] \end{array}$$
considering the small attitude error due to AHRS. Therefore, the limit is taken for Equation (4). $C_{b}^{n}$ can be expressed as follows:(5)$$C_{b}^{n}=C_{b^{e}}^{n}\left[\begin{array}{ccc} 1 & -\delta \psi & \delta \theta \\ \delta \psi & 1 & -\delta \varphi \\ -\delta \theta & \delta \varphi & 1 \end{array}\right]$$

Let $C_{b^{e}}^{n}$ be written as ${\hat{C}}_{b}^{n}$, and substitute Equation (4) into Equation (5).
(6)$$\begin{array}{cc} {\overset{\cdot }{v}}_{n} & ={\hat{C}}_{b}^{n}\left[\begin{array}{ccc} 1 & -\delta \psi & \delta \theta \\ \delta \psi & 1 & -\delta \varphi \\ -\delta \theta & \delta \varphi & 1 \end{array}\right]f_{b}-g_{n}+C_{b}^{n}\delta f\\ & ={\hat{C}}_{b}^{n}f_{b}+{\hat{C}}_{b}^{n}\left[\begin{array}{ccc} 0 & -\delta \psi & \delta \theta \\ \delta \psi & 0 & -\delta \varphi \\ -\delta \theta & \delta \varphi & 0 \end{array}\right]f_{b}-g_{n}+C_{b}^{n}\delta f\\ & ={\hat{C}}_{b}^{n}f_{b}+{\hat{C}}_{b}^{n}\left[-f_{b}x\right]\left[\begin{array}{c} \delta \varphi \\ \delta \theta \\ \delta \psi \end{array}\right]-g_{n}+C_{b}^{n}\delta f \end{array}$$

Equation (6) finally establishes the UAV’s velocity differential equation in the navigation coordinate system based on AHRS.
(7)$${\dot{v}}_{n}={\hat{C}}_{b}^{n}\left[-f_{b}x\right]\left[\begin{array}{c} \delta \varphi \\ \delta \theta \\ \delta \psi \end{array}\right]+{\hat{C}}_{b}^{n}f_{b}-g_{n}+C_{b}^{n}\delta f$$

Here ${\hat{C}}_{b}^{n}$ is obtained from the output of the AHRS system, and δφ, δψ, and δθ represent the attitude measurement errors of the AHRS.

The differential equation for the relative velocity of the UAV can be obtained by the differential equation for the velocity of a single UAV. Assuming the differential equation for the velocity of a single lead UAV is equal to:(8)$${\dot{v}}_{l}={\hat{C}}_{bl}^{l}\left[-f_{bl}x\right]\left[\begin{array}{c} \delta \varphi_{l}\\ \delta \theta_{l}\\ \delta \psi_{l} \end{array}\right]+{\hat{C}}_{bl}^{l}f_{bl}-g_{n}+C_{b}^{l}\delta f_{l}$$

Similarly, the differential equation for the velocity of the follow UAV is equal to:(9)$${\dot{v}}_{f}={\hat{C}}_{bf}^{f}\left[-f_{bf}x\right]\left[\begin{array}{c} \delta \varphi_{f}\\ \delta \theta_{f}\\ \delta \psi_{f} \end{array}\right]+{\hat{C}}_{bf}^{f}f_{bf}-g_{n}+C_{b}^{f}\delta f_{f}$$

In the study of UAV formation flight, the dynamic changes in the relative positions of the two aircraft lead to corresponding changes in their respective navigation coordinate systems. Simply using the velocity differential equations of the leader and the wingman UAVs would overlook the differences between the two navigation coordinate systems, potentially introducing computational errors. To accurately describe the acceleration states in the relative velocity differential equation, this research constructs a coordinate transformation matrix from the navigation system to the Earth-Centered, Earth-Fixed (ECEF) system based on the position information provided by the single aircraft navigation system. Through this coordinate transformation matrix, the accelerations of the leader and the wingman UAVs, denoted as ${\dot{v}}_{l}$ and ${\dot{v}}_{f}$, are converted to the ECEF system. Finally, the relative velocity differential equation is obtained through calculation.
(10)$$\begin{array}{ll} {\dot{v}}_{r} & =C_{f}^{e}{\dot{v}}_{f}-C_{l}^{e}{\dot{v}}_{l}\\ & =C_{bf}^{e}f_{bf}+C_{bf}^{e}\left[-f_{bf}x\right]\left[\begin{array}{l} \delta \varphi_{f}\\ \delta \theta_{f}\\ \delta \psi_{f} \end{array}\right]-C_{f}^{e}g_{n}+C_{bf}^{e}\delta f_{f}-C_{bl}^{e}f_{bl}-C_{bl}^{e}\left[-f_{bl}x\right]\left[\begin{array}{l} \delta \varphi_{l}\\ \delta \theta_{l}\\ \delta \psi_{l} \end{array}\right]+C_{l}^{e}g_{n}-C_{bl}^{e}\delta f_{l} \end{array}$$

Continuing from the previous discussion, since $C_{l}^{e}g_{n}$ − $C_{f}^{e}g_{n}$ is a relatively small value, Equation (10) can be simplified as follows:(11)$${\dot{v}}_{r}=C_{bf}^{e}f_{bf}+C_{bf}^{e}\left[-f_{bf}x\right]\left[\begin{array}{l} \delta \varphi_{f}\\ \delta \theta_{f}\\ \delta \psi_{f} \end{array}\right]+C_{bf}^{e}\delta f_{f}-C_{bl}^{e}f_{bl}-C_{bl}^{e}\left[-f_{bl}x\right]\left[\begin{array}{l} \delta \varphi_{l}\\ \delta \theta_{l}\\ \delta \psi_{l} \end{array}\right]-C_{bl}^{e}\delta f_{l}$$
where $C_{bf}^{e}$ and $C_{bl}^{e}$ are obtained from the longitude and latitude information provided by the individual navigation systems of the drones. $f_{bf}$ and $f_{bl}$ represent the acceleration outputs of each drone, while $\delta \psi_{f}$, $\delta \theta_{f}$, $\delta \varphi_{f}$, $\delta \psi_{l}$, $\delta \theta_{l}$, and $\delta \varphi_{l}$ denote the attitude output errors of each AHRS system. Additionally, $\delta f_{f}$ and $\delta f_{l}$ are the output errors of each accelerometer.

In the modeling of the differential equation for relative position, considering the complexity of the system and the brief duration of the filtering cycle, this equation is simplified to a constant velocity motion model. Additionally, addressing the issue of random offset in accelerometers, this study describes it as a first-order Markov process. Based on this assumption, and considering that the error models for the three axes have similar characteristics, it can be expressed as:(12)$$\left[\begin{array}{l} \delta {\dot{f}}_{x}\\ \delta {\dot{f}}_{y}\\ \delta {\dot{f}}_{z} \end{array}\right]=\left[\begin{array}{lll} -1/T_{ax} & & \\ & -1/T_{ay} & \\ & & -1/T_{az} \end{array}\right]\left[\begin{array}{l} \delta f_{x}\\ \delta f_{y}\\ \delta f_{z} \end{array}\right]+w(t)$$
where *T_ax_*, *T_ay_*, and *T_az_* are the correlation times, and ***w***(t) is a Gaussian white noise with a mean of zero.

## 3. Methods

This paper introduces an adaptive filtering method. It is based on system innovation and formulates the adaptive judgment rule. It uses measurement equations and values to design the optimal objective function. Additionally, it incorporates the TDO algorithm into the filtering method. The method can improve the system’s stability and accuracy, and at the same time, it is easy to realize.

### 3.1. Design of Adaptive UAV Navigation Method Based on AHRS

This paper is based on the system model proposed in Section 2. At the same time, the effect of partial sensor measurement noise is considered under the UAV equipped with multiple sensors. Combined with the EKF algorithm, an AHRS-based UAV adaptive navigation method is proposed. The example is the UAV *l*, and the design of the method is shown in Figure 2.

The design treats the AHRS mentioned in Section 2 as a black box. Using the attitude information it provides and the multi-sensor observations of the UAV, a two-stage filtering structure is designed. The TDO algorithm is also used to correct the state that deviates from the real motion to improve the accuracy and stability of the method.

The design of each stage of the filtering method is described in the following sections.

#### 3.1.1. Combined AHRS/GNSS Filtering

Taking UAV *l* as an example, the filtering method is formed using a tight combination of GNSS and AHRS, as shown in Figure 3. The main function of this filtering in this paper’s method is twofold: (1) closed-loop error correction of the gyroscope and accelerometer through the equation of state modeling of the sensor error; and (2) use of the internal composition of the AHRS to form a Strapdown INS. Its output ***X****_l_* of absolute UAV navigation location information is used for relative navigation state filtering.

The combined local filter equation of state is as follows:(13)$$X_{k}^{e}=F_{k/k-1}^{e}X_{k-1}^{e}+G_{k-1}^{e}W_{k-1}^{e}$$
where the state variables $X_{k}^{e}$ are selected as:(14)$$X_{k}^{e}={\left[\Phi_{E},\Phi_{N},\Phi_{U},\delta v_{E},\delta v_{N},\delta v_{U},\delta L,\delta \lambda ,\delta h,\varepsilon_{cgx},\varepsilon_{cgy},\varepsilon_{cgz},\varepsilon_{rgx},\varepsilon_{rgy},\varepsilon_{rgz},\varepsilon_{rax},\varepsilon_{ray},\varepsilon_{raz},\delta s_{u},\delta s_{ru}\right]}^{T}$$

This $X_{k}^{e}$ consists of the error; the first 6 dimensions are the local filter output parameter error, which includes the 3-dimensional inertial navigation platform error angle, the 3-dimensional velocity error, and the 3-dimensional position error. The 10th to 15th state variables, respectively, are gyroscope three-axis constant drift error, and three-axis random drift error. The 16th to 18th dimensions are accelerometer three-axis random drift errors. The last two dimensions are the distance error due to the equivalent clock error, and the distance error due to the equivalent clock frequency error.

Based on the error analysis of the Strapdown inertial navigation system, several error equations can be obtained [31].

The platform error angle equation is expressed as:(15)$$\dot{\Phi }=\delta {\bar{\omega }}_{\mathrm{in}}^{n}+\Phi \times {\bar{\omega }}_{\mathrm{in}}^{n}-\varepsilon^{n}$$
where, $\varepsilon^{n}$ denotes the gyroscope instrument error, and $\delta {\bar{\omega }}_{\mathrm{in}}^{n}$ and $\Phi \times {\bar{\omega }}_{\mathrm{in}}^{n}$ are the gyroscope drift caused by the rotation motion of the Earth and the UAV’s motion.

The velocity error equation is given as:(16)$$\delta \dot{v}=\Phi \times f^{n}+\nabla^{n}-\left(2\delta \omega_{\mathrm{ie}}^{n}+\delta \omega_{\mathrm{en}}^{n}\right)\times v^{n}-\left(2\omega_{\mathrm{ie}}^{n}+\omega_{\mathrm{en}}^{n}\right)\times \delta v+\delta g$$
where δg is the accelerometer error, $\omega_{\mathrm{ie}}^{n}$ is the Earth’s rotational velocity, and $\omega_{\mathrm{en}}^{n}$ is the projection of the UAV N-coordinate system onto the N-coordinate system relative to the E-coordinate system.

The position error equation is represented as:(17)$$\{\begin{array}{c} \delta \overset{\cdot }{L}=\frac{v_{U}}{{\left(R_{M}+h\right)}^{2}}\delta h+\frac{1}{R_{M}+h}\delta v_{N}\\ \delta \dot{\lambda }=\frac{v_{E}\tan L}{\left(R_{N}+h\right)\cos L}\delta L-\frac{v_{E}}{{\left(R_{N}+h\right)}^{2}\cos L}\delta h+\frac{1}{\left(R_{N}+h\right)\cos L}\delta v_{E}\\ \delta \dot{h}=\delta v_{U} \end{array}$$
where *R* is the radius of the Earth, about 6317 KM, *R_M_* = (1 − 2f + 3fsin2L)Re, *R_N_* = (1 + fsin2L)Re, and f = 1/298.257.

Accelerometers and gyroscopes typically include mounting errors, scale factor errors, and instrumentation random errors, which are usually colored noise. In this section, these physical errors are uniformly considered as comprehensive random errors.

Let the gyro drift error consist of scale factor error, zero-bias error, random constant, and white noise error.
(18)$$\varepsilon =\varepsilon_{bias}+\varepsilon_{g}+w_{g}$$

Assuming that the zero-bias error and the scale factor error are modeled as a first-order Markov process, the following is obtained:(19)$$\{\begin{array}{c} {\dot{\varepsilon }}_{bias=0}\\ {\dot{\varepsilon }}_{g}=-\frac{1}{T_{g}}\varepsilon_{g}+w_{r} \end{array}$$
where *T*_g_ is the gyroscope correlation time.

Let the accelerometer error model consist only of a first-order Markov process and assume that the error model is the same for the 3 axial directions of the accelerometer. The same reasoning is obtained:(20)$${\dot{\varepsilon }}_{a}=-\frac{1}{T_{a}}\varepsilon_{a}+w_{a}$$
where *T*_a_ is the accelerometer correlation time.

The GNSS receiver correlation error is modeled as:(21)$$\{\begin{array}{c} \delta {\overset{\cdot }{s}}_{u}=\delta s_{\mathrm{ru}}+w_{\mathrm{tu}}\\ \hat{\delta }{\overset{\cdot }{s}}_{\mathrm{ru}}=-\beta_{\mathrm{tru}}\hat{\delta }{\overset{\cdot }{s}}_{\mathrm{ru}}+w_{\mathrm{tru}} \end{array}$$
where $\delta {\overset{\cdot }{s}}_{u}$ and $\hat{\delta }{\overset{\cdot }{s}}_{\mathrm{ru}}$ are the distance error due to the receiver equivalent clock error and the distance error due to the equivalent clock frequency error, respectively.

By associating Equations (15)–(21), the state transfer matrix $F_{k/k-1}^{e}$ and the noise coefficient matrix $G_{k-1}^{e}$ can be obtained.

In designing the measurement equations for combining the local filters, this section adopts a tight combination approach: i.e., instead of directly using the GNSS-solved positioning information, mathematical modeling is performed based on the pseudorange observed by the GNSS receiver.

Using the pseudorange value $\rho_{li}$ obtained from the position calculation and the receiver measurement pseudorange value $\rho_{gi}$, the observation equation is obtained by difference processing [32]:(22)$$\Delta \rho_{i}=\rho_{li}-\rho_{gi}=\frac{\partial \rho_{\mathrm{Ii}}}{\partial x}\delta x+\frac{\partial \rho_{\mathrm{Ii}}}{\partial y}\delta y+\frac{\partial \rho_{\mathrm{Ii}}}{\partial z}\delta z+\delta t_{u}+v_{\rho i}$$
where $\rho_{li}$ is the calculated value of the pseudorange and $\rho_{gi}$ is the measured value of the pseudorange. δx, δy, δy are the position errors in the E-coordinate system.

According to the local filter state variables, the position error is output in the N-coordinate system. Therefore, it is necessary to convert the position error in the E-coordinate system to the N-coordinate system. The conversion equation is expressed as:(23)$$\{\begin{array}{c} \delta x=\delta h\cos L\cos\lambda -\left(R_{N}+h\right)\sin L\cos\lambda \delta L-\left(R_{N}+h\right)\cos L\sin\lambda \delta \lambda \\ \delta y=\delta h\cos L\sin\lambda -\left(R_{N}+h\right)\sin L\sin\lambda \delta L+\left(R_{N}+h\right)\cos L\cos\lambda \delta \lambda \\ \delta z=\delta h\sin L+\left[R_{N}{\left(1-f\right)}^{2}+h\right]\cos L\delta L \end{array}$$

Bringing Equation (23) into Equation (22), the equation for the pseudorange differential measurement is given as:(24)$$Z_{k}^{e}=\delta \rho =H_{k}^{e}X_{k}^{e}+V_{k}^{e}$$
where $H_{k}^{e}$ is the pseudorange observation equation and $V_{k}^{e}$ is the pseudorange measurement noise.

Ultimately, the error data for each sensor can be obtained by EKF. This error data is then used to calibrate the accelerometer and gyroscope outputs in the AHRS. Also, using the accelerometer and gyroscope data, the local filter can provide the UAV with absolute positioning information. This information is valuable in improving the overall accuracy of the relative positioning method.

#### 3.1.2. Relative Navigation State Filtering

Relative navigation state filtering is designed based on the model presented in Section 2. Set the state variables and state equations for this filtering method according to Equations (11) and (12).
(25)$$\begin{array}{c} X^{r}=[v_{r\_x},v_{r\_y},v_{r\_z},r_{x},r_{y},r_{z},\delta f_{l\_x},\delta f_{l\_y},\delta f_{l\_z},\delta f_{f\_x},\delta f_{f\_y},\delta f_{f\_z}]\\ X^{r}(t)=F_{r}(t)X^{r}+B(t)U_{r}+G(t)W(t) \end{array}$$
where $v_{r}$ is the relative velocity vector, ***r*** is the relative position vector, and $\delta f_{l}$, $\delta f_{f}$ are the ratio error vectors of UAV *l* and UAV *f*, respectively.

Next, the state transfer matrix can be represented as:(26)$$F_{r}(t)={\left[\begin{array}{cccc} 0_{3\times 3} & 0_{3\times 3} & -C_{bl}^{e} & C_{bf}^{e}\\ 0_{3\times 3} & I_{3\times 3} & 0_{3\times 3} & 0_{3\times 3}\\ 0_{3\times 3} & 0_{3\times 3} & \frac{-1}{T_{a}} & 0_{3\times 3}\\ 0_{3\times 3} & 0_{3\times 3} & 0_{3\times 3} & \frac{-1}{T_{a}} \end{array}\right]}_{12\times 12}$$

The system control variables are assumed to be:(27)$$U_{r}=C_{bf}^{e}f_{bf}-C_{bl}^{e}f_{bl}$$

The control variable coefficient matrix is expressed as:(28)$$B(t)=\left[\begin{array}{c} I_{3\times 3}\\ 0_{9\times 3} \end{array}\right]$$

Set the system process noise W(t) to:(29)$$\left[\begin{array}{cccccccccccc} \delta \varphi_{l} & \delta \theta_{l} & \delta \psi_{l} & \delta \varphi_{f} & \delta \theta_{f} & \delta \psi_{f} & W_{al} & W_{al} & W_{al} & W_{af} & W_{af} & W_{af} \end{array}\right]$$

The noise factor matrix G(t) is denoted as:(30)$$\left[\begin{array}{ccc} C_{bl}^{e}\left[f_{bl}x\right] & 0_{3\times 3} & 0_{3\times 6}\\ 0_{3\times 3} & C_{bf}^{e}\left[-f_{bf}x\right] & 0_{3\times 6}\\ 0_{3\times 6} & I_{3\times 3} & 0_{3\times 3}\\ 0_{3\times 6} & 0_{3\times 3} & I_{3\times 3} \end{array}\right]$$

The discretized equation of state is obtained by discretizing Equation (25).
(31)$$X_{k}^{r-}=F_{k|k-1}X_{k-1}^{r+}+BU_{k}+G_{k-1}W_{k-1}$$

UWB is not only used as a communication module to provide data interaction for the navigation system, but it also provides ranging functions. Its observation equation and observation matrix are denoted as [33]:(32)$$\begin{array}{l} r_{UWB}={\left\|r\right\|}_{2}+n_{UWB}\\ H_{UWB}=[0_{1\times 3}\;\frac{\partial r_{UWB}}{\partial r}\;0_{1\times 6}] \end{array}$$

GNSS not only forms a filter with AHRS but also forms DGNSS by differential means, eliminating common errors present in the pseudorange. The observation matrix and observation equation for the double difference between pseudorange and pseudorange rates are expressed as [1]:(33)$$\begin{array}{l} \Delta \nabla \rho_{lf}^{S_{1}S_{i}}=\left(e^{S_{i}}-e^{S_{1}}\right)r+n_{lf}^{S_{1}S_{i}}\;\;\;\;,i=2,\ldots ,8\\ \Delta \nabla {\dot{\rho }}_{lf}^{S_{1}S_{i}}=\left(e^{S_{i}}-e^{S_{1}}\right)v+\left({\dot{e}}^{S_{i}}-{\dot{e}}^{S_{1}}\right)r+n_{lf\_rate}^{S_{1}S_{i}}\;,i=2,\ldots ,8\\ H_{dd}=\left[\begin{array}{ccc} e^{Sj}-e^{Si} & {\dot{e}}^{S_{j}}-{\dot{e}}^{S_{1}} & 0_{1\times 6}\\ \vdots & \vdots & \vdots \\ 0_{1\times 3} & e^{S_{j}}-e^{S_{1}} & 0_{1\times 6}\\ \vdots & \vdots & \vdots \end{array}\right] \end{array}$$
where $e^{S_{i}}$ is the direction cosine vector from the midpoint to the satellite S_i_ between the two UAVs, ${\dot{e}}^{S_{i}}$ is the change rate of the direction cosine vector from the midpoint to the satellite S_i_ between the two UAVs, r is the relative position vector between the UAVs, v is the relative velocity vector between the UAVs, $n_{lf}^{S_{1}S_{i}}$ is the noise that exists in the pseudorange bi-differential computation, $n_{lf_{-}\mathrm{rate}\;}^{S_{1}S_{i}}$ is the noise that exists in the pseudorange bi-differential rate computation, and both kinds of noise belong to Gaussian white noise.

The AHRS/GNSS filtering method outputs the absolute navigation information of the UAV along with the closed-loop correction of the sensor. Based on this, the relative velocity and relative position of the UAV can be obtained. The measurement matrix of this value is expressed as:(34)$$\begin{array}{l} r_{abs}=r+n_{abs}\\ v_{abs}=v+n_{abs\_rate}\\ H_{\mathrm{abs}}=\left[\begin{array}{cccc} I_{3} & 0_{1\times 3} & 0_{1\times 3} & 0_{1\times 3}\\ 0_{1\times 3} & I_{3} & 0_{1\times 3} & 0_{1\times 3} \end{array}\right] \end{array}$$

Associating Equation (31) to Equation (33), the measurement model for relative navigation state filtering is obtained as:(35)$$\begin{array}{l} Z_{k}^{r}=H_{k}^{r}X_{k}^{r-}+V^{r}\\ H_{k}^{r}={\left[\begin{array}{lll} H_{dd} & H_{\mathrm{abs}} & H_{UWB} \end{array}\right]}^{T} \end{array}$$
where $V^{r}$ is the measurement noise vector.

After outputting the relative navigation state $X_{k}^{r+}$ using the EKF, the value is subjected to an adaptive judgment. If it deviates from the true motion then it is corrected using the TDO algorithm.

### 3.2. TDO Algorithm Principle and Flow

TDO is an optimization algorithm inspired by nature [34]. This algorithm mimics the behavioral patterns of the pouched Tasmanian devil when searching for food and is used to solve complex optimization problems. There are two strategies within the TDO algorithm: attacking live prey or feeding on the carrion of the animal.

The TDO algorithm is a population-based stochastic algorithm whose search subject is the Tasmanian devil. So, in conjunction with the scenario of this paper, the initialization of the pouched Tasmanian devil population is obtained from the relative navigated system a posteriori state with the system state covariance matrix, and each member of the pouched Tasmanian devil represents a set of navigated state quantities. Each member of the TDO population is a searcher on the solution space, so that the TDO navigated state member population can be described as a matrix:(36)$$X={\left[\begin{array}{c} X_{1}\\ \vdots \\ X_{i}\\ \vdots \\ X_{N} \end{array}\right]}_{N\times m}={\left[\begin{array}{ccccc} x_{1,1} & \cdots & x_{1,j} & \cdots & x_{1,m}\\ \vdots & \ddots & \vdots & \ddots & \vdots \\ x_{i,1} & \cdots & x_{i,j} & \cdots & x_{i,m}\\ \vdots & \ddots & \vdots & \ddots & \vdots \\ x_{N,1} & \cdots & x_{N,j} & \cdots & x_{N,m} \end{array}\right]}_{N\times m}$$
where ***X*** represents the whole navigation state population, ***X****_i_* represents the *i*_th_ population member, and *X*_i,j_ represents the *j*_th_ state quantity in member *i*. Through this matrix, it can be seen that N is the number of navigation state population members, and m is the dimension of the system to solve the state quantity.

Bringing ***X*** in Equation (36) into the objective function *F*(*x*) yields a vector ***F*** consisting of function values for each navigation state of the candidate.
(37)$$F={\left[\begin{array}{c} F_{1}\\ \vdots \\ F_{i}\\ \vdots \\ F_{N} \end{array}\right]}_{N\times 1}={\left[\begin{array}{c} F\left(X_{1}\right)\\ \vdots \\ F\left(X_{i}\right)\\ \vdots \\ F\left(X_{N}\right) \end{array}\right]}_{N\times 1}$$
where ***F*** is the vector composed of objective function values and ***F****_i_* is the objective function value obtained for the *i*_th_ navigation state. Each objective function value demonstrates the quality of the corresponding candidate solution. The candidate solution that can compute the optimal value of the objective function is considered as the optimal member of the population. In each iteration, the optimal member of the population is updated based on the new value.

#### 3.2.1. Strategy 1: Carrion Eater Strategy

The strategy of navigating the state as a “carrion eater” is mathematically modeled by Equations (38)–(40). In the TDO design, for each member it is assumed that the positions of other population members in the search space are “carrion” positions. Random selection of carrion locations is modeled in Equation (38).
(38)$$C_{i}=X_{k},\;i=1,2,\ldots ,N,k\in \{1,2,\ldots ,N|k\neq i\}$$
where ***C****_i_* represents the carrion location chosen by the *i*_th_ member of the population, and ***X****_k_* is the location of all members except the *i*_th_ member.

The new position of the member in the search space is calculated based on the selection of carrion; in this strategy, the member will move towards the carrion if the carrion objective function value is better and vice versa, then the member will move away from the carrion, which is modeled by Equation (39).
(39)$$x_{i,j}^{\mathrm{new},S1\;}=\{\begin{array}{ll} x_{i,j}+r\cdot \left(c_{i,j}-I\cdot x_{i,j}\right), & F_{C_{i}}<F_{i}\\ x_{i,j}+r\cdot \left(x_{i,j}-c_{i,j}\right), & \;\mathrm{else}\; \end{array}$$

Equation (39) will calculate the new position of the member of the navigation state; if this objective function value is better in this new position then *X_i_* will be updated, otherwise *X_i_* will remain in its original position.
(40)$$X_{i}=\{\begin{array}{ll} X_{i}^{\mathrm{new},S1}, & F_{i}^{\mathrm{new},S1}<F_{i}\\ X_{i}, & \;\mathrm{else}\; \end{array}$$
where X_*i*_^new,S1^ is the new value calculated by the “carrion strategy”, *X*_*i*,*j*_^new,S1^ is the *j*_th_ element in ***X***_*i*_^new,S1^, *F*_*i*_ is the objective function value, *F*_*i*_^new,S1^ is the objective function value of the carrion position, *r* is a random value between 0 and 1, and *I* is 1 or 2.

#### 3.2.2. Strategy 2: Predator Strategy

The second strategy of the TDO algorithm is the predator strategy, where the member’s behavior during predation has two phases: in the first phase, the prey is selected by searching the area; then, the second phase will keep approaching the selected prey to attack. The selection of prey in the first phase is similar to the modeling of Carrion Eater Strategy
(41)$$P_{i}=X_{k},\;i=1,2,\ldots ,N,k\in \{1,2,\ldots ,N|k\neq i\}$$
where ***P***_i_ is the prey chosen by the *i*_th_ member, ***X****_k_* is any other population member except *i*, and *k* is a random number between 1 and N.

After determining the position of the prey, the new position of the member is calculated, and if the objective function value of the chosen prey is better then the member moves towards it, and vice versa it moves away from that position
(42)$$x_{i,j}^{\mathrm{new},S2}=\{\begin{array}{ll} x_{i,j}+r\cdot \left(p_{i,j}-I\cdot x_{i,j}\right), & F_{P_{i}}<F_{i}\\ x_{i,j}+r\cdot \left(x_{i,j}-p_{i,j}\right), & \;\mathrm{else}\; \end{array}$$
(43)$$X_{i}=\{\begin{array}{ll} \begin{array}{ll} X_{i}^{\mathrm{new},S2}, & F_{i}^{\mathrm{new},S2}<F_{i}\\ X_{i}, & \;\mathrm{else} \end{array} & \end{array}$$
where *X*_*i*_^new,S2^ is the new value calculated in the first phase of the predator strategy, *X*_*i*,*j*_^new,S2^ is the *j*_th_ element of *X*_*i*_^new,S2^, ***F***_*i*_^new,S2^ is the objective function value of the new value, $F_{i}$ is the objective function value of the position of the prey, *r* is a random value between 0 and 1, and *I* is 1 or 2.

In order to simulate the process of members chasing the prey near the prey location, the prey location obtained in the first stage is used as the center of the chase and a search radius is set to represent the chase range, with the search radius computed as:(44)$$R_{TDO}=0.01\left(1-\frac{t}{T}\right)$$
where *t* is the current iteration number and *T* is the maximum iteration number of the algorithm. The mathematical modeling of the members pursuing the prey in the region is as follows:(45)$$x_{i,j}^{new}=x_{i,j}+(2r-1)\cdot R_{TDO}\cdot x_{i,j}$$
(46)$$X_{i}=\{\begin{array}{ll} X_{i}^{\mathrm{new}\;}, & F_{i}^{\mathrm{new}\;}<F_{i}\\ X_{i}, & \;\mathrm{else}\; \end{array}$$
where *R*_*TDO*_ is the radius of prey determined through Equation (44), *X*_*i*_^new^ is the location of prey determined in the second phase of the predator strategy, *F*_*i*_^new^ is the value of *X*_*i*_^new^ after it is brought into the objective function, and *X*_*i*,*j*_^new^ is the *j*_th_ element of *X*_*i*_^new^.

The TDO algorithm flowchart is shown in Figure 4.

### 3.3. Adaptive Judgment Rule Design

EKF is a commonly used algorithm for data fusion, and the principle of the algorithm will not be repeated in this paper; the algorithm calculates the optimal value of the state quantities need to use the innovation, which is denoted as:(47)$$e=Z_{k}^{r}-H_{k}^{r}X_{k}^{r-}$$
where $Z_{k}^{r}$ denotes the sensor observation value at the current moment, $H_{k}^{r}$ denotes the observation matrix, and $X_{k}^{r-}$ denotes the a priori state quantity calculated by the state equation. When the UAV motion model does not match the system state equation or the UAV has a sudden change in motion, the measurement prediction value obtained after the nonlinear measurement function introduced by the state equation will have a large deviation, which is propagated to the innovation ***e*** by Equation (24).

Since the Kalman filtering algorithm utilizes the innovation for a posteriori estimation of state quantities, in order to make the filtering results more accurate, it is necessary to judge the innovation ***e***. The innovation ***e*** can reflect the error between the current measurement prediction and the real measurement, and it can also show the “size” of the deviation of the current state value from the actual motion state. The innovation e can reflect the error between the current measurement prediction and the real measurement, and it can also show the “size” of the current state value deviation from the actual motion state, so we can use the innovation to design the adaptive filter judgment threshold.

When designing the adaptive filtering judgment threshold, it is considered that the measurement noise of each sensor is Gaussian white noise. Therefore, when each sensor works normally, the measurement covariance matrix ***R*** can be used to judge whether the innovation *e* is out of the error range of the measurement. Assuming that the current measurement value $Z_{i}^{r}$ is the center of the judgment range, and because the components on the diagonal of the measurement covariance matrix represent the variance of the corresponding observation value, $\sqrt{R_{i}}$ (the *i* component on the diagonal of the covariance matrix) is taken as the radius of the judgment range, and the error within the range belongs to the normal range. When the absolute value of the innovation exceeds this range, it is necessary to start correcting the state values by the algorithm.

Combined with the use of UAV multi-sensors, it can be seen that in the filtering algorithm process there will be more than one measurement value, and the measurement value accuracy of different sensors is not the same; in order to more reasonably use the measurement value to determine whether the state quantity needs to be corrected, it is necessary to carry out appropriate preprocessing for $Z_{i}^{r}$ and ***e***. Through the elements on the diagonal of ***R*** we can know each kind of measured value error size, and through the preprocessing we hope that in the adaptive correction judgment, the sensor with high accuracy can play a greater role, so the proposed preprocessing formula is
(48)$$a_{j}=\frac{\sum_{i=1}^{k}\sqrt{R_{i}}-\sqrt{R_{j}}}{\sum_{i=1}^{k}\sqrt{R_{i}}}$$
(49)$$\bar{e}=\sum_{j=1}^{k}a_{j}|e_{j}|$$
(50)$$\lambda =\sum_{j=1}^{k}a_{j}\sqrt{R_{j}}$$
where *k* is the number of UAV sensors, $\bar{e}$ is the preprocessing value obtained from the current innovation, and λ is the adaptive judgment threshold. According to Equations (48)–(50), it can be seen that when the accuracy of the sensors is higher the coefficient *a*_*j*_ in the calculation of the judgment threshold is larger.

In practical application, the following adaptive judgment rule is proposed: through $\bar{e}$ with λ to recognize the abnormal situation of state quantity, when $\bar{e}$ exceeds λ, the relative navigation state will be corrected by using the TDO algorithm, through which the process ensures that the algorithm can adapt to the relative motion state changes of the UAV, so as to improve its stability in the dynamic environment.

### 3.4. Objective Function Setting and Performance Analysis of TDO

With the development of multi-source sensors, the sensor measurement accuracy is getting higher and higher, so Equation (47) is set as the objective function. And because the measurement equations between the sensors are all linearly independent, it can be seen that the rank of the ***H*** matrix is greater than the dimension of the relative navigation state volume.
(51)$$Rank(H_{k}^{r})>D(X_{r})$$

Therefore, the optimal solution can be found by setting Equation (51) as the objective function.
(52)$$F_{obj}=e=Z_{k}^{r}-H_{k}^{r}X_{k}^{r-}$$

According to the principle analysis of the TDO algorithm in the previous section, it can be known that the time complexity of TDO is mainly related to *N*, *m*, and *T*. The algorithm is processed in a loop and the time complexity is O(*N***m***T*).

The performance of the TDO algorithm is further analyzed using Equation (52) as the objective function. Sensor measurements and navigation data are derived from analogue values. The advantages of TDO are verified by comparing GA and PSO. The optimization results are shown in Figure 5.

The TDO algorithm has better performance than traditional optimization algorithms in high-dimensional optimization problems like UAV state solving.

### 3.5. TDO Adaptive Kalman Filtering Algorithm Flow

Based on the relative motion state equation, adaptive judgment rule, and TDO algorithm proposed above, combination with the extended Kalman filter algorithm can give the adaptive Kalman filter algorithm flow.

Initialize the number of TDO iterations ***t***, the maximum number of TDO iterations ***T***, the number of TDO populations ***N***, the initial covariance matrix of the system ***P***_0_, the initial state of the system ***X***_0_, and the measurement noise covariance matrix ***R***:

Step 1. The system state and covariance matrix are predicted in one step, i.e.,
$$\{\begin{array}{c} X_{k}^{r-}=F_{k,k-1}X_{k-1}^{r+}+BU_{k}\\ P_{k}^{-}=F_{k,k-1}P_{k-1}^{+}F_{k,k-1}^{T}+G_{k-1}Q_{k-1}G_{k-1}^{T} \end{array};$$

Step 2. The gain matrix ***K***_*k*_ is computed from the measurement noise covariance matrix ***R***, i.e.,
$$K_{k}=P_{k}^{-}H_{k}^{rT}{\left[H_{k}^{r}P_{k}^{-}H_{k}^{rT}+R_{k}\right]}^{-1};$$

Step 3. Calculate the innovation ***e*** and combine it with ***K***_*k*_ using Equation (47) for optimal estimation of the relative navigation state, i.e.,
$$X_{k}^{r+}=X_{k}^{r-}+K_{k}e;$$

Step 4. Update the system state covariance matrix, i.e.,
$$P_{k}^{+}=\left(I-K_{k}H_{k}^{r}\right)P_{k}^{-};$$

Step 5. Use the new innovation ***e*** and ***R*** through Equation (48) to Equation (50) to obtain the adaptive filtering threshold λ and $\bar{e}$;

Step 6. If $\bar{e}$ < λ, then the algorithm ends outputting $X_{k}^{r+}$ with $P_{k}^{+}$, otherwise go to step 7;

Step 7. If t < T, generate N population members by arraying control system state upper and lower bounds in accordance with the objective function Equation (52) with the number of populations N. Then, use Equations (36)–(46) for TDO algorithm loop iteration solution, otherwise the algorithm ends outputting the optimal solution of the relative navigational state $X_{k}^{r+}$.

## 4. Results

### 4.1. Simulation Initial Conditions

The UAV *l* and UAV *f* adopt a close-range accompanying flight mode, and their flight trajectories are shown in Figure 6. It is assumed that each aircraft adopts the GPS/INS tight combination positioning method, and real-time relative navigation is carried out during the flight, and the flight duration is 600 s.

Table 1 lists the parameter settings in the relative navigation method proposed in this paper. The number of satellites, the maximum number of iterations of the TDO algorithm, and the number of TDO population members.

Table 2 lists the parameters of each sensor device used for the UAV with each measurement noise. Both UAVs are equipped with IMU, satellite navigation receiver, UWB sensors, pseudorange, and pseudorange rate, and the UWB measurement noise is Gaussian white noise.

### 4.2. Simulation Results and Analysis

We aim to verify that the relative motion state equation derived based on AHRS can more accurately describe the relative motion of the UAV compared to the CA/CV motion equation, and at the same time, adaptive Kalman filtering can reduce the system error in the case of mismatch between the state equation and the motion state; therefore, this paper compares the results of relative navigation in three configurations:AHRS+TDO adaptive Kalman filtering method: the method proposed in this paper, using the relative navigation equation of state derived based on AHRS, using UWB, relative differential, and dual positioning differential data as observation data;CA/CV equation of state: traditional relative motion equation of state, using observation data consistent with method 1;AHRS: only the relative motion equation of state derived based on AHRS is used, using observation data consistent with method 1;

The cumulative distribution function of the relative position errors of the three methods is first compared. As shown in Figure 7, according to the distribution and slope of the CDF curve, the error of method 1 is concentrated near 0 and has the largest slope. The relative position error–CDF curves for Method 2 and Method 3 are similar.

The following text analyses the stability of three methods based on the standard deviation of the position errors in the E-coordinate system’s X, Y, and Z directions. Figure 8 shows that Method 1 is the most stable method due to its adaptive method based on the AHRS motion state equation. Method 2, on the other hand, is unable to correct the state value that deviates from the actual motion, resulting in less stability than Method 1. Finally, Method 3, which employs the traditional positioning technique, has the lowest stability among the three methods.

Table 3, along with Figure 9, Figure 10 and Figure 11, presents the simulation results of three methods under the RMSE performance index. Firstly, in terms of relative position results, using the RMSE in X, Y, and Z directions as the comparison term, the AHRS+TDO adaptive Kalman filtering method demonstrated the highest accuracy. The second highest accuracy was obtained by using only the relative motion equation based on AHRS. The traditional CA/CV motion model method showed the worst relative navigation performance.

The relative velocity error CDF curves for Method 1 and Method 3, are shown in Figure 12. The error distribution of method 1 is significantly better than that of method 3. This is due to the accurate description of the relative motion by the AHRS-based motion model proposed in this paper.

Method 1 and method 3’s relative velocity error STDs are shown in Figure 13. Simulation results demonstrate that the AHRS-based adaptive filtering method has better stability.

Upon comparing the relative velocity error, the algorithm proposed in this paper has significantly enhanced the relative velocity navigation accuracy in comparison to the traditional CA/CV model. This can be observed in Figure 14, Figure 15 and Figure 16. The accuracy of relative navigation in the X, Y, and Z directions has improved by 3.5, 3.3, and 3 times, respectively. For more specific data, please refer to Table 4.

This paper proposes a UAV navigation method based on AHRS. The system model uses the relative motion equation to enhance the method’s accuracy. The multi-stage filter structure corrects the sensor error. Additionally, the method utilizes adaptive judgment rules to correct the state that deviates from the real motion, which further improves the method’s stability and accuracy.

## 5. Conclusions

In this paper, relative navigation methods for UAVs are investigated. Aiming to solve the problem of inaccurate description of the relative navigation state by the traditional CA/CV motion model, the state equation for relative navigation is derived based on the aerial position information provided by AHRS, and a state equation more suitable for relative navigation scenarios is proposed. Considering the navigation accuracy degradation problem caused by the mismatch between the state equation and the motion state during the navigation process, the adaptive judgment rule based on innovation judgment is proposed, the TDO objective function is designed by using the measurement information and the measurement equation, and, finally, the state quantities that deviate from the real motion are solved by the TDO algorithm. The simulation results show that the relative navigation method proposed in this paper can effectively improve the accuracy of the navigation system, effectively reduce the relative position error under the same sensor conditions compared with the traditional CA/CV model, and simultaneously reduce the relative speed error substantially, and it is suitable for the application scenarios where the UAVs have high requirements for relative navigation accuracy and robustness such as in formation.

## Figures and Tables

**Figure 1 sensors-24-02518-f001:**
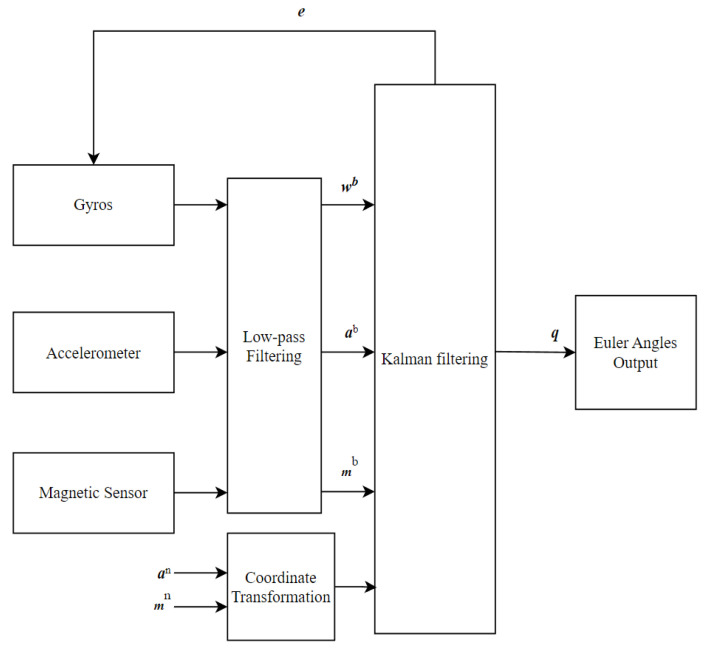
Attitude computing system architecture for AHRS.

**Figure 2 sensors-24-02518-f002:**
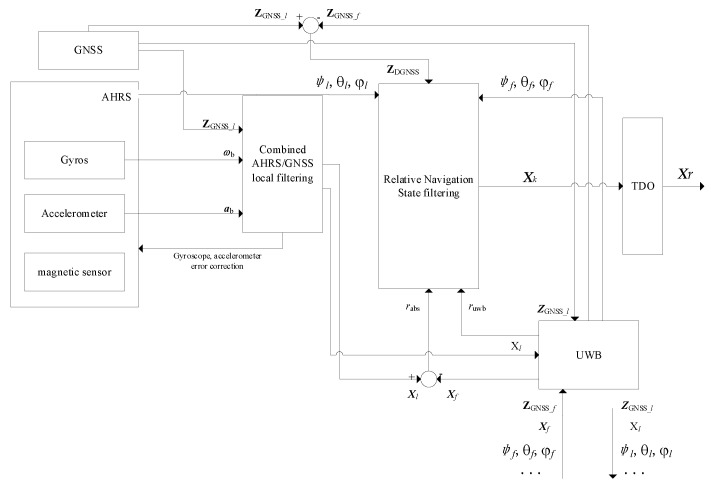
Design of adaptive UAV navigation method based on AHRS.

**Figure 3 sensors-24-02518-f003:**
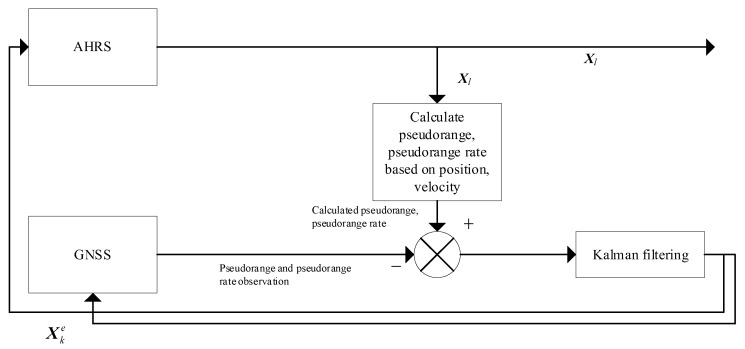
Combined AHRS/GNSS filtering.

**Figure 4 sensors-24-02518-f004:**
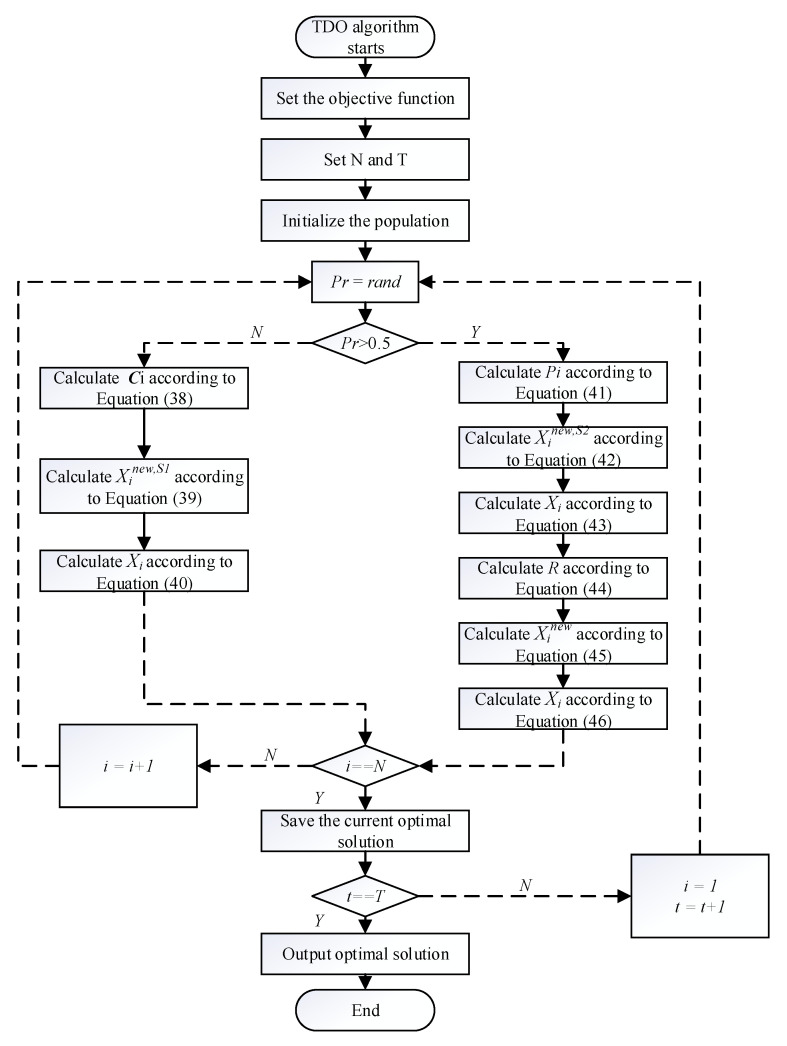
Flowchart of TDO algorithm.

**Figure 5 sensors-24-02518-f005:**
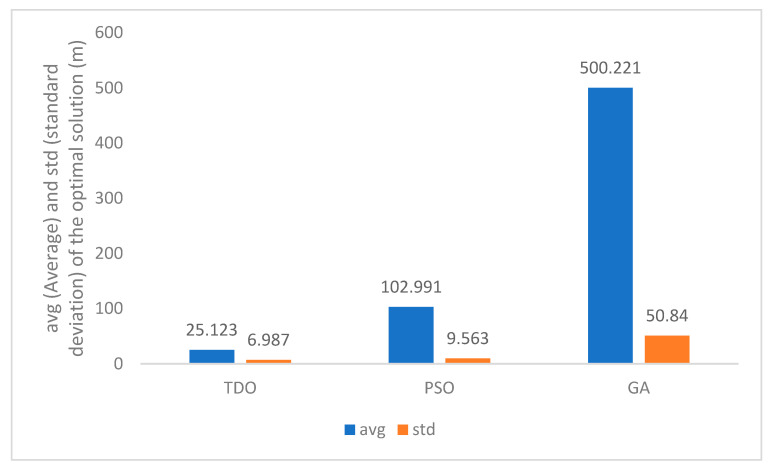
Optimization results of TDO and competitor algorithms.

**Figure 6 sensors-24-02518-f006:**
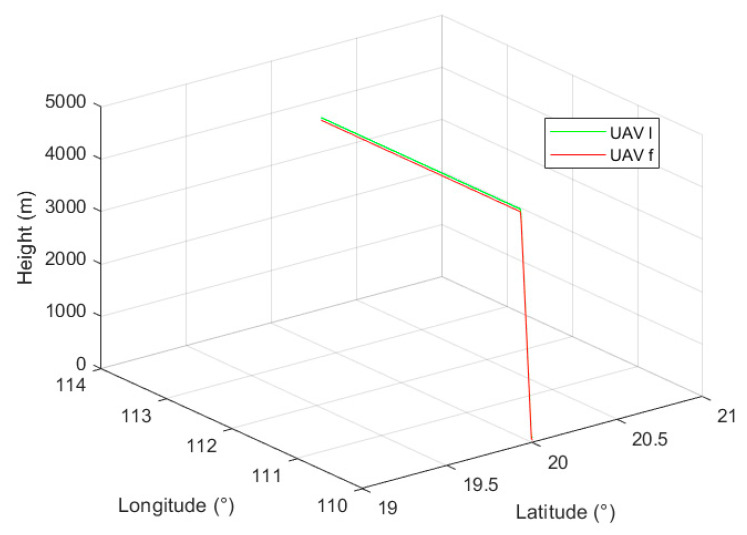
UAV *l* and UAV *f* flight trace.

**Figure 7 sensors-24-02518-f007:**
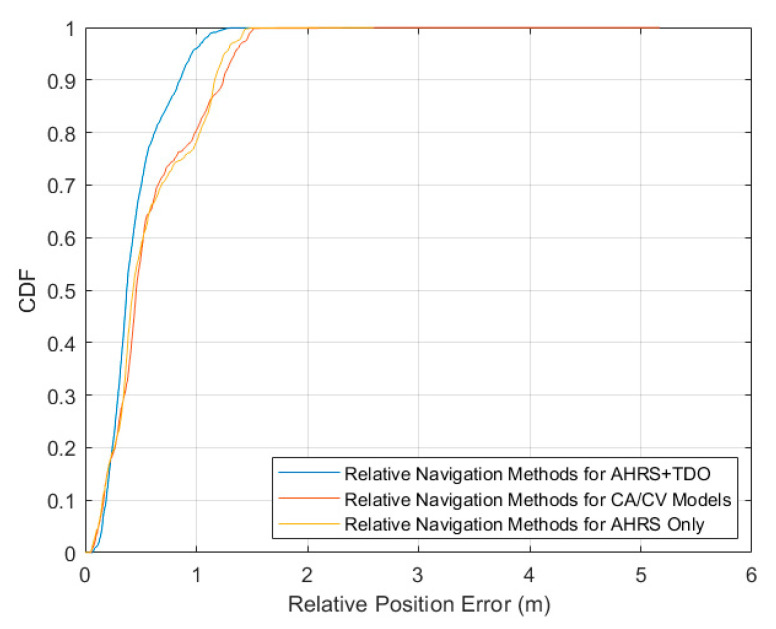
Relative position error versus CDF.

**Figure 8 sensors-24-02518-f008:**
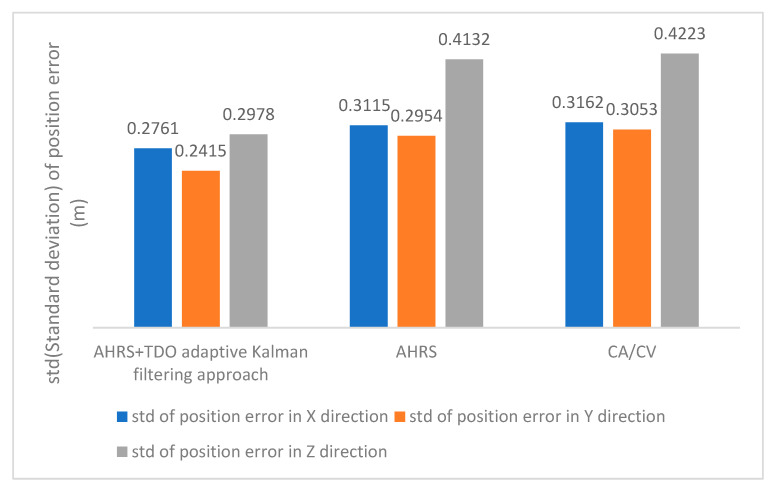
Relative position error in the direction of X, Y, and Z axes std.

**Figure 9 sensors-24-02518-f009:**
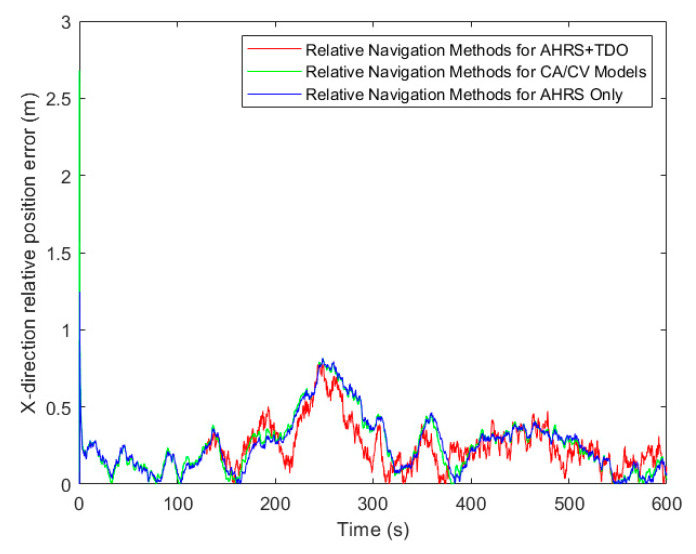
Comparison of relative position error in X-direction.

**Figure 10 sensors-24-02518-f010:**
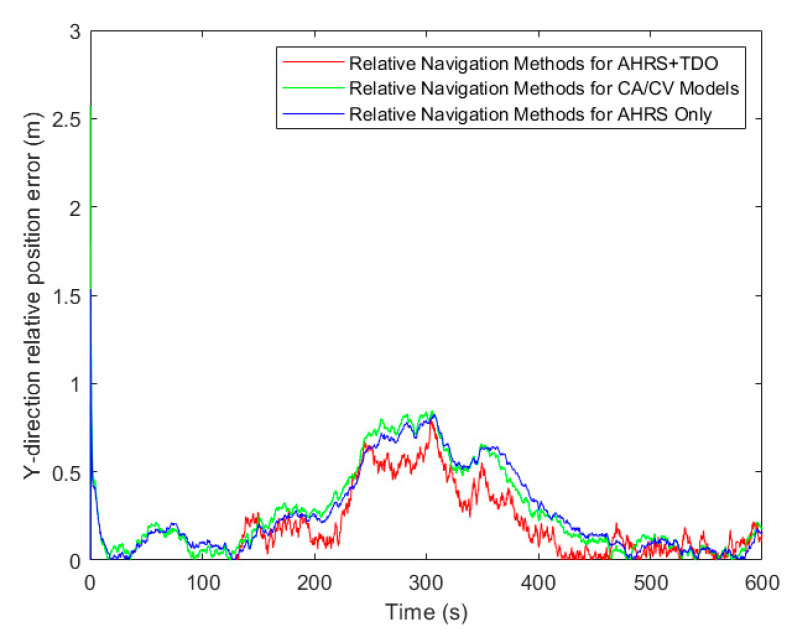
Comparison of relative position error in Y-direction.

**Figure 11 sensors-24-02518-f011:**
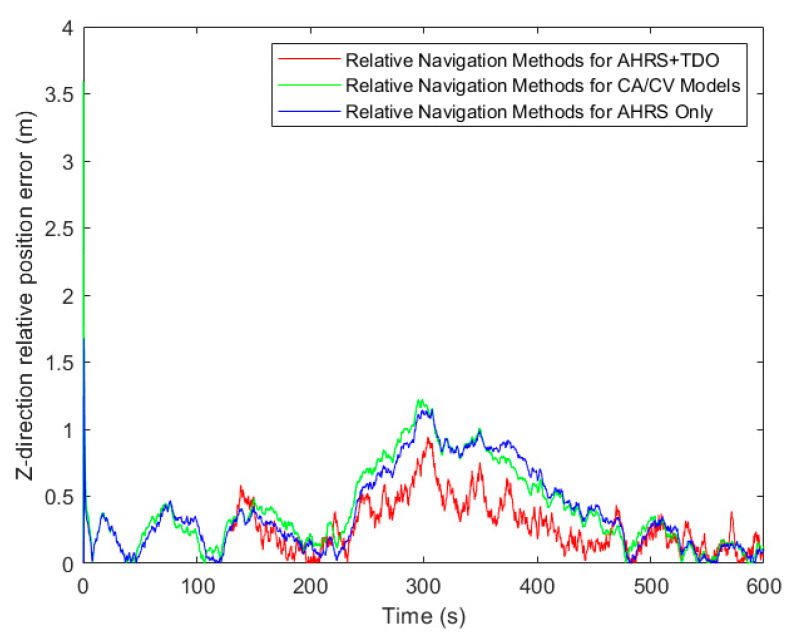
Comparison of relative position error in Z-direction.

**Figure 12 sensors-24-02518-f012:**
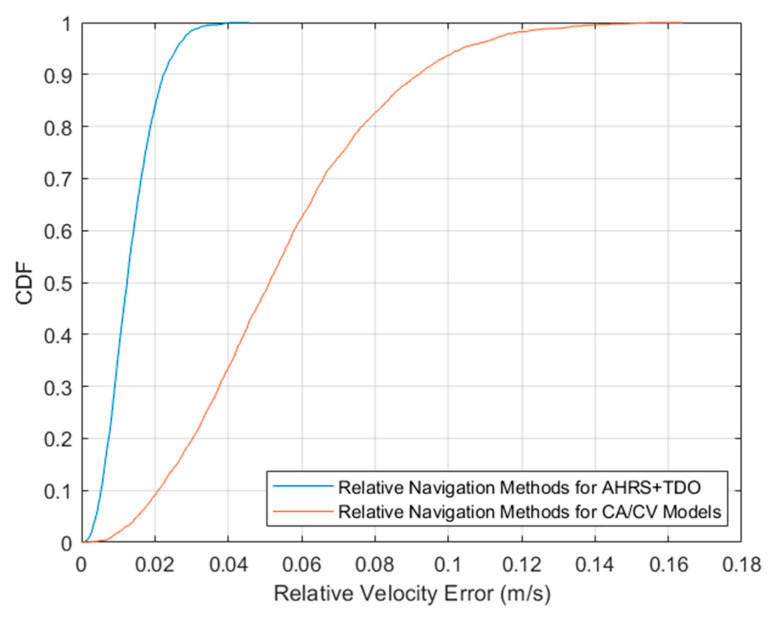
Relative velocity error CDF.

**Figure 13 sensors-24-02518-f013:**
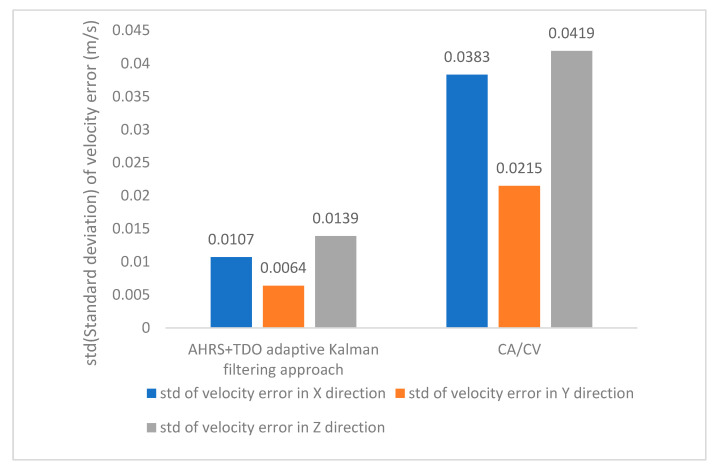
Relative velocity error in the direction of X, Y, and Z axes std.

**Figure 14 sensors-24-02518-f014:**
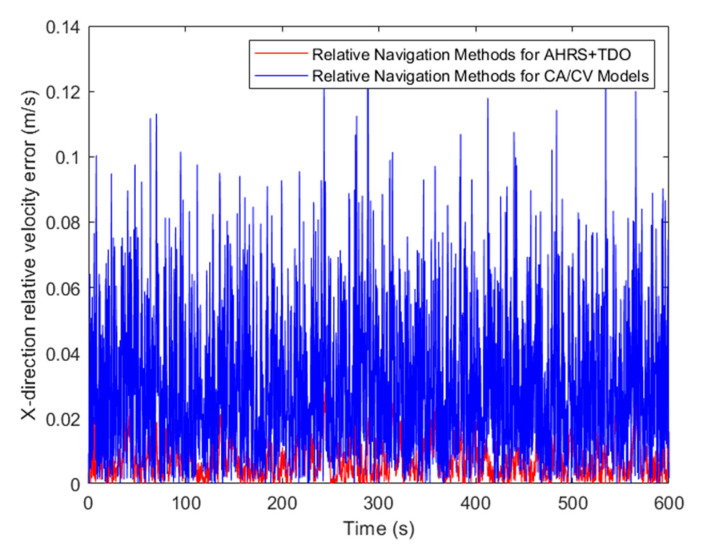
Comparison of relative velocity error in X-direction.

**Figure 15 sensors-24-02518-f015:**
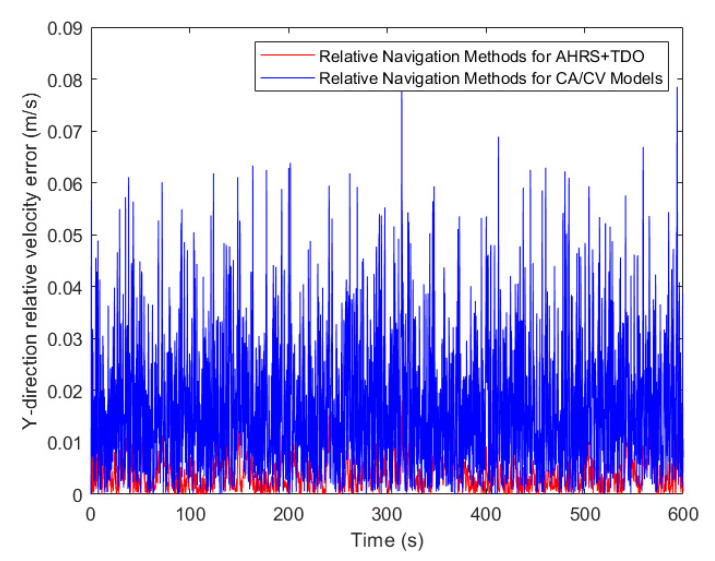
Comparison of relative velocity error in Y-direction.

**Figure 16 sensors-24-02518-f016:**
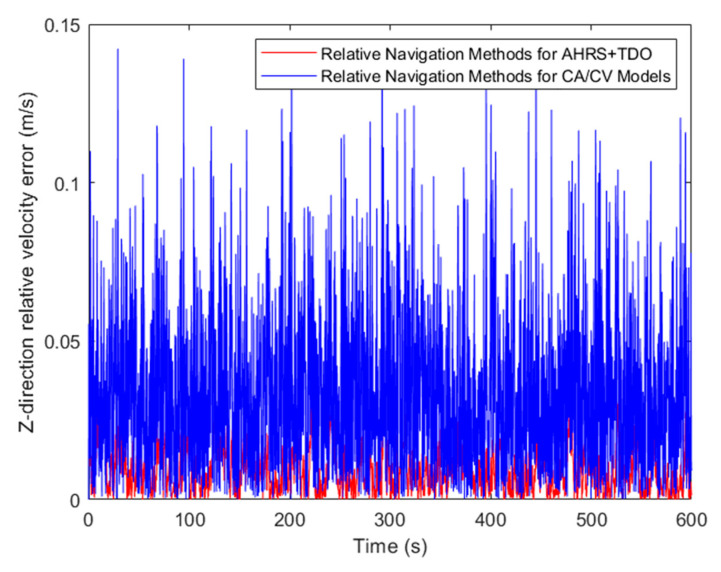
Comparison of relative velocity error in Z-direction.

**Table 1 sensors-24-02518-t001:** Relative navigation method Parameters.

Parameters	Meaning	Value
*N* _s_	Number of available satellites	8
*N*	Number of TDO stock members	30
*T*	Maximum number of TDO iterations	800

**Table 2 sensors-24-02518-t002:** Sensor Configuration and Measurement Noise Settings.

Sensors	Parameters	Value
Gyros	Constant Drift	0.1 (°)/h
	White noise error	0.1 (°)/h
	First-order Markov random noise	0.1 (°)/h
	First-order Markov correlation time	3600 s
Accelerometer	First-order Markov random noise	0.01 g
	First-order Markov correlation time	0.03 m/s
GPS	Pseudorange error	3 m
	Pseudorange rate error	0.03 m/s
UWB	Ranging noise	0.15 m
	Crystal Error Scaling Factor	1 × 10^−3^

**Table 3 sensors-24-02518-t003:** Relative position error comparison.

Relative Position Error	RMSE(m)		
	AHRS+TDO Adaptive Kalman Filtering Approach	AHRS	CA/CV
X-direction	0.27857	0.31351	0.31852
Y-direction	0.28461	0.35852	0.36996
Z-direction	0.34116	0.49446	0.50831

**Table 4 sensors-24-02518-t004:** Relative speed error comparison.

Relative Speed Error	RMSE (m/s)	
	AHRS+TDO Adaptive Kalman Filtering Approach	CA/CV
X-direction	0.01076	0.03826
Y-direction	0.00644	0.02151
Z-direction	0.01393	0.04194

## Data Availability

Data are contained within the article.

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
