# Peer review of "Adaptive UAV Navigation Method Based on AHRS"

_sensors, 2024, doi:10.3390/s24082518_

Round 1
Reviewer 1 Report
Comments and Suggestions for Authors
In this paper, the authors report an adaptive UAV relative navigation method based on the UAV attitude information provided. By integrating the extended Kalman filter for relative state estimation and employing an adaptive decision rule designed using the innovation of the filter update phase, the proposed method recalculates motion states deviating from the actual motion using the Tasmanian Devil Optimization (TDO) algorithm.
As a reviewer, I think that the subject is consistent with the scope of journal. However, Experimental techniques discussed in this manuscript lack novelty and data analysis is relatively straightforward. The manuscript reads mostly like a test report with some perhaps noteworthy conclusions in the applications of GNC.
Thank you
Reviewer 2 Report
Comments and Suggestions for Authors
The paper describes a UAV navigation method based on adaptive state estimation. The main paper contribution is the utilization of the TDO algorithm in state filtering. The contribution is obvious and supported by simulation results, however, the qualitative aspect of the presented method in comparison with analogs remains open. Moreover, the reviewer still has additional questions and comments:
1. The list of optimization methods is extensive and the advantage of this application is not clear. In addition, there is no description of the algorithm's performance, although in UAV systems real-time is critical given the high operating velocities.
2. The source of formulas (2)-(4) (and others below) that describe the accelerometer measurement model is not indicated. Typically, the accelerometer and gyroscope measurement model contain a component that describes the bias. Does this work consider bias?
3. What does the transition to a differential equation for velocity give in the accelerometer measurement model?
4. What innovation do the authors mean – line 91 “By utilizing the innovation obtained during the Kalman filtering process”
5. Please clarify the meaning of the phrase “system, thereby replacing the concept of the traditional mathematical platform.”, line 124.
6. In lines 16-17, it is not the Kalman filter is integrated, but its output.
7. In Fig. 1 there is no error signal signature, although in the text the notation is highlighted. Moreover, the "Error Calculation" block output should be one of the "Error Correction" inputs by meaning.
8. In line 174, first point, is it meant to introduce the UAV dynamics equations directly? It is also possible to derive the measurement equations in more detail, taking into account Markov noise and other factors.
9. In line 179, "in Section 1" should be used instead of "in part 1". It is unclear exactly what part of what is meant.
10. In lines 307-308, the notations H(t) are duplicated.
11. Fig. 3 lacks a legend for the graphs.
12. The graphs in Fig. 4 are not quite informative. Perhaps a larger scale should be used to emphasize the estimation process itself.
13. It is not recommended to start a sentence with a reference, for example, line 45, 55, etc.
With the above comments, the article requires minor revision.
Comments on the Quality of English LanguageMinor editing of English language required
Reviewer 3 Report
Comments and Suggestions for Authors
In this manuscript, the authors presented an adaptive UAV relative navigation method, which is based on the UAV attitude information provided by Attitude and Heading Reference System (AHRS). By integrating the extended Kalman filter for relative state estimation and employing an adaptive decision rule designed using the innovation of the filter update phase, the proposed method recalculates motion states deviating from the actual motion using the Tasmanian Devil Optimization (TDO) algorithm. In general, I think this manuscript might be considered as potential publication in “Sensors” after addressing some issues. Please see the comments to authors.
1. The format of this paper should be carefully double-checked. Some minor mistakes can be found in the manuscript, for example, there is a lot of white space on line 300.
2. Table 4 should be written on the same page as far as possible.
Comments on the Quality of English LanguageReasonable English description
Round 2
Reviewer 1 Report
Comments and Suggestions for Authors
The main novelty of presented work is not mentioned
